# Non-Structural Protein-W61 as a Novel Target in Severe Fever with Thrombocytopenia Syndrome Virus (SFTSV): An In-Vitro and In-Silico Study on Protein-Protein Interactions with Nucleoprotein and Viral Replication

**DOI:** 10.3390/v15091963

**Published:** 2023-09-20

**Authors:** Ji-Young Park, Chandran Sivasankar, Perumalraja Kirthika, Dhamodharan Prabhu, John Hwa Lee

**Affiliations:** 1Department of Veterinary Public Health, College of Veterinary Medicine, Jeonbuk National University, Iksan 54596, Republic of Koreapkirthika@mayo.edu (P.K.); 2Department of Biochemistry and Molecular Biology, Mayo Clinic, Rochester, MN 55905, USA; 3Centre for Drug Discovery, Karpagam Academy of Higher Education, Coimbatore 641021, India; dprabhubio@gmail.com

**Keywords:** SFTSV, viral protein, protein-protein interaction, in-vitro and in-silico analysis

## Abstract

The non-structural protein (NSs) and nucleoprotein (NP) of the severe fever with thrombocytopenia syndrome virus (SFTSV) encoded by the S segment are crucial for viral pathogenesis. They reside in viroplasm-like structures (VLS), but their interaction and their significance in viral propagation remain unclear. Here, we investigated the significance of the association between NSs and NP during viral infection through in-silico and in-vitro analyses. Through in-silico analysis, three possible binding sites were predicted, at positions C6S (Cystein at 6th position to Serine), W61Y (Tryptophan 61st to Tyrosine), and S207T (Serine 207th to Threonine), three mutants of NSs were developed by site-directed mutagenesis and tested for NP interaction by co-immunoprecipitation. NSsW61Y failed to interact with the nucleoprotein, which was substantiated by the conformational changes observed in the structural analyses. Additionally, molecular docking analysis corroborated that the NSW61Y mutant protein does not interact well compared to wild-type NSs. Over-expression of wild-type NSs in HeLa cells increased the SFTSV replication by five folds, but NSsW61Y exhibited 1.9-folds less viral replication than wild-type. We demonstrated that the W61Y alteration was implicated in the reduction of NSs-NP interaction and viral replication. Thus, the present study identified a critical NSs site, which could be targeted for development of therapeutic regimens against SFTSV.

## 1. Introduction

Severe fever with thrombocytopenia syndrome virus (SFTSV) belonging to *Bunyavirales* [1] is considered one of the most prevalent medically important tick-borne human pathogens that cause serious illnesses, including a range of fevers, viral hemorrhagic diseases, encephalitis, and meningitis [2,3,4]. As the range of their host vector expands, there is an increased risk of transmission, and the number of countries where infections have been reported is increasing [5,6,7]. The SFTSV genome consists of three negative-strand RNA segments: the L segment encoding RNA dependent RNA polymerase (RdRp) involved in viral replication; the M segment encoding glycoprotein precursors (Gn and Gc); and the S segment encoding nucleoprotein (NP) and nonstructural proteins (NSs) [8]. NP is essential for transcription and replication of viral life cycles and viral genomes, forming ribonucleoprotein complexes. NSs have been reported to play an essential role in SFTSV propagation and can form viroplasm-like structures (VLSs) in infected and transfected cells. NSs-formed VLSs are associated with lipid droplets (LDs), and SFTSV viral replication can be affected by lipid metabolism and immune signaling pathways [8,9]. NSs are associated with viral NP and RNA and may be involved in viral RNA replication [9]. Furthermore, it has been reported that NSs can act as a skeletal protein supporting SFTSV RNA replication, as replication intermediates of viral RNA in the form of NP and dsRNA have been found in VLS [10,11]. The role of NS proteins and their significance in SFTSV RNA replication are unknown. As studies are underway on the therapy associated with emerging viruses such as SFTSV, discovering the role of NSs in virulence can accelerate the research and development of therapeutic regimens.

With this background, the present study is aimed at deciphering the key binding sites of NS and its significance in interaction with NP through computational structural analysis and in vitro protein-protein interaction analysis. Three mutants corresponding to site-directed mutations of NS were developed and screened to identify the functional sites of NS-NP interaction through in-silico and immuno-precipitation technology. Furthermore, the role of key binding sites of NSs and their implications in viral replication were demonstrated for the first time by in-silico and in vitro study. The key findings of the study will be pivotal in the research on therapeutics against SFTSV.

## 2. Materials and Methods

### 2.1. Cell Lines and Virus

Human embryonic kidney cells (HEK293T, ATCC CRL3216™), HeLa cells (ATCC CCL-2), and Vero E6 cells (ATCC CCL-81) were maintained in Dulbecco’s Modified Eagle’s Medium, supplemented with 10% fetal bovine serum and 1% antibiotic-antimycotic (Gibco, Thermo Fisher Scientific, Inc., Waltham, MA, USA). Then, SFTS virus strain KADGH (NCCP 43261) was purchased from the National Culture Collection for Pathogens (NCCP, Osong, Chungbuk, Korea). SFTSV virus was propagated with Vero E6 cells for amplification, and viral titers were determined by FAID_50_ assay and stored at −80 °C until further use. All experiments using SFTSV were performed in the BSL-3 laboratory facility according to guidelines of Jeonbuk National University biosafety committee (JBNU2019-01-001-001) at the Korea Zoonosis Research Institute (KOZRI, Iksan, Jeonbuk, Korea).

### 2.2. Accession Numbers

The S segments from 12 SFTSV (Korean isolates) strains from the NCBI GenBank were selected and assembled using CLC Genomics Workbench version 7.5.2 (CLC Bio, Cambridge, MA, USA) to generate the consensus sequences. These strains were CB1 (KY789439), CB2 (KY789440), CB3 (KY789441), KACNH3 (KP663745), KADGH (KJ739545), KAGBH5 (KP663739), KAGBH6 (KP663742), KAGNH (KU507555), KAGNH4 (KJ739557), KAGWH3 (KP663736), KAJNH2 (KJ739563), and KASJH (KP663733). Consensus sequences that have been aligned are presented in Appendix A.

### 2.3. Construction of the DNA Plasmids

Bacterial strains, plasmids, and primers used in this study are listed in Appendix A. The full-length NS and NP consensus sequences were synthesized by Bioneer (Daejeon, Korea). The NS and NP genes were amplified from the synthesized genes with indicated primers and cloned into pcDNA3.1-CMV for eukaryotic cell expression under the cytomegalovirus (CMV) promoter. The pcDNA3.1-NSs mutant plasmids were generated using an EZ change™ site-directed mutagenesis kit (Enzynomics, Daejeon, Korea), using pcDNA3.1-NSs constructed as a template. Finally, the mutations in the final constructs were verified by DNA sequencing.

### 2.4. Protein Expression and Purification

The NSs, NSsW61Y, and NP genes were cloned into a 6× His-PET28a (+) vector system, and recombinant plasmids were transformed into *E. coli* BL21 DE3 (Novagen, Madison, WI, USA) for protein expression. The expression of each gene was induced with isopropyl-β-d-thiogalactoside (IPTG, Sigma-Aldrich, Saint Louis, MO, USA), and each protein fraction was purified using NI-NTA column chromatography (TAKARA, Shiga, Japan). The purified protein was quantified by Bradford assay and confirmed by Western blot analysis using anti-his-Tag antibodies (Santa Cruz Biotechnology, Inc., Dallas, TX, USA) [12]. The purified proteins were then stored at −80 °C until further use.

### 2.5. Antibody Production

All animal experiments were performed with the approval of the Jeonbuk National University Animal Ethics Committee (JBNU-2019-00284). Specific pathogen-free female New Zealand white rabbits were purchased from Samtako (Osan, Kyungkido, Korea), housed in an animal facility, and provided with antibiotic-free food and water ad libitum. Rabbits were immunized with NSs and NP purified proteins in an equal volume of Freund’s complete adjuvant (Sigma-Aldrich) via the subcutaneous route. Two weeks after initial immunization, booster inoculations were performed subcutaneously using Freund’s incomplete adjuvant (Sigma-Aldrich).

### 2.6. Circular Dichroism (CD) Spectroscopy

CD analysis was performed on structures of WT and mutant proteins of NSs at the National Instrumentation Center for Environmental Management (NICEM, Seoul National University, Seoul, Korea) using a Chirascan CD spectrometer (Applied Photophysics, Leatherhead, UK). Data were collected between 190 and 260 nm through the appropriate buffer and solvent background subtraction. The data of the residual ellipticity rate (deg cm^2^ mo^−1^) obtained from the analysis were processed by the online-based server “CAPITO: A CD analysis and plotting tool” (https://data.nmr.uni-jena.de/capito/index.php, accessed on 7 October 2021) [13].

### 2.7. In-Silico Analysis

The 3D structures of NP and NSs were predicted using protein homology/analogy recognition engine (Phyre^2^) [14] and illustrated with Pymol (Schrödinger, LLC; Version 1.2r3pre) and AlphaFold [15]. The Ramachandran plot was analyzed through a PROCHECK server to measure the quality of and validate the 3D structure [16]. The protein preparation was carried out with the help of docking software Schrödinger and predicted the binding site with larger volume and area by using CASTp server. Protein grid generation was done by using the default parameters of van der Waals scaling factor.

### 2.8. Immunofluorescence Assay (IFA) and Fluorescence Active Infectious Dose (FAID_50_) Assay

HeLa or HEK293T cells were transfected with pcDNA3.1 NSs or pcDNA3.1 NSsW61Y using lipofectamine 3000 (Invitrogen, Thermo Fisher Scientific, Inc., Waltham, MA, USA) The transfected cells were subjected to SFTSV viral infection at MOI of 0.5 for 24 h and 48 h. The cell supernatant of virus-infected HeLa cells was collected at 48 h and placed on Vero E6 cells in a 96-well plate at 37 °C for 1 h to determine the viral load. Cells were added to DMEM-supplemented 4% FBS, and the plates were incubated at 37 °C in 5% CO_2_ for five days. The cells were washed with 1× phosphate-buffered saline (PBS) three times and fixed with 80% cold acetone for 10 min at −20 °C. Next, cells were washed three times with 1× PBS and blocked with 5% bovine serum albumin (BSA, GeneAll, Seoul, Republic of Korea) in 1× PBS for 1 h at 37 °C. The cells were then incubated with NSs and NP hyperimmune serum from mice or rabbit overnight at 4 °C. The following day, cells were washed with 1× phosphate-buffered saline with, 0.1% Tween (1× PBST) and incubated with secondary Alexa 488 anti-Rabbit IgG and Alexa 647 anti-mouse IgG (Invitrogen, Thermo Fisher Scientific, Inc.) at 1:2000 dilution for 1 h at 37 °C. Next, cells were washed and counterstained with 4′,6-diamidino-2-phenylindole (DAPI) (Sigma-Aldrich) for 10 min. Finally, the cells were observed using a fluorescence microscope (Leica Biosystems, Nussloch, Germany).

### 2.9. Immunoprecipitation (IP) Assay and Immunoblot Analysis

To examine the binding of NP to exogenous NSs or NSsW61Y derivatives, HEK293T cells were seeded in six-well plates (2 × 10^6^ cells/well), and pcDNA3.1-NP plasmid was co-transfected with each NS and NSsW61Y expression plasmid using polyethyleneimine (PEI). At 48 h post-transfection, cells were washed with 1× PBS and lysed using a lysis buffer (25 mM Tris-HCl, 150 mM NaCl, 1 mM EDTA, and 1% Triton X-100) containing a protease inhibitor cocktail (Invitrogen) in ice. After centrifugation at 13,000× *g* for 15 min at 4 °C, the supernatants were incubated at 4 °C overnight with 10 µL of SFTSV-NP rabbit hyper-immune serum and protein A/G agarose beads (Santa Cruz Biotechnology, Dallas, Texas,). The following day, the cell lysate mixture was precipitated with protein A/G plus agarose beads. Agarose beads were then washed three times with washing buffer (1× PBS with 0.5% Triton X-100, pH 7.4). The precipitated beads were eluted by boiling in 5 × SDS sample buffer and subjected to sodium dodecyl sulfate (SDS)-polyacrylamide gel electrophoresis (PAGE) and transferred to a PVDF membrane (Millipore, MA, USA). Membranes were treated with SFTSV-NP or NSs hyper-immune serum after being blocked with 5% skim milk. After three washes in tris-buffered saline with tween 20 (TBST) the membrane was subsequently treated with horseradish peroxidase–conjugated goat anti-rabbit antibody (Southern Biotechnology, Birmingham, AL, USA), developed with Amersham ECL detection reagents (Cytiva, WA, USA) and imaged using the Amersham Image Quant 800 imaging system (Cytiva).

### 2.10. Quantitative Real-Time PCR

SFTSV was inoculated into HeLa cells transfected with expression plasmid for each NP, NS and NSW61Y at MOI of 0.5 and then culture supernatant harvested for RNA isolation at 24 h and 48 h post-infection. Viral RNA from supernatant infected with SFTSV was isolated using an AccuPrep^®^ viral RNA extraction kit (Bioneer, Daejeon, Korea), and first-strand cDNA was synthesized using a cDNA synthesis kit (Elpis Biotech, Daejeon, Korea) according to the manufacturer’s instructions. Quantitative real-time PCR was performed with EzAMP^TM^ qPCR 2x Master Mix SYBR (Elpis Biotech) following the manufacturer’s procedures. The relative SFTSV mRNA expression of the S segment was evaluated by genome copy number, experimental and biological triplicates were subjected to calculate the viral copy numbers.

### 2.11. Statistical Analysis

All data analysis was conducted in Excel and Prism software 8.0 (GraphPad Software, Inc., San Diego, CA, USA). Statistical analysis was performed using Student’s *t*-test, where *p* ≤ 0.05 was considered statistically significant. Significance levels were defined as ** *p* ≤ 0.01, *** *p* < 0.001, and **** *p* < 0.0001.

## 3. Results

### 3.1. Construction Characterization of the NP, NSs, and NS Mutant Proteins

To determine the similarity between genomic sequences, we retrieved sequences of 12 SFTSV Korean isolates (from 2012 to 2016) that were available in GenBank and generated a multiple sequence alignment based S segment. We found that the NSs and NP from SFTSV strains belonging to genotypes A, B, and F had a relatively high level of homology (>90%) at the nucleotide level (Appendix A). At the same time, we produced consensus sequences of each NS and NP using CLC Genomics Workbench software version 7.5.2. According to a previous study, NS interacts with NP in VLSs [17]. To identify possible binding regions between NSs and NP, we performed 3D structure prediction using AlphaFold software [15]. The NSs structure is composed of 15 alpha helices and two beta sheets. There were two distinct domains observed in the NS protein and the majority of the secondary structural elements are in the N-terminal domain. The N-terminal domain is fully made of 12 alpha helices (α1–α12), whereas the C-terminal domain contains three alpha helices (α13–α15) and two beat sheets (β1 and β2). The phi and psi distribution of the amino acids of the modelled NSs and NP protein in ramachandran plot depicted that more than 90% of the residues were distributed in the allowed regions, the 3D modelled structure is reliable for further docking studies (Appendix A). The accurate binding sites prediction of the protein are the crucial factor in determining the best possible protein-protein interactions. The CASTp server predicted the binding site with larger volume and area of 3034.46 Å3 and 3625.78 Å2. In the 3D model, three binding sites were predicted at amino acid residues from 1 to 32, 56 to 82, and 207 to 237 a.a for NP, which are highlighted in orange in Figure 1A. In the first replacement, CYS6 (cysteine at the sixth position) was replaced with SER (serine); in the second replacement, TRP61 (tryptophan at 61st position) was changed to TYR (tyrosine), and in the third replacement, SER207 (serine at 207th position) was replaced with TEROH (threonine) through site-directed mutation. Preliminarily, the substitutions at all three positions (individually) in NSs failed to interact with the NP as simulated by ClusPro (Figure 1B).

### 3.2. Effects of NS Mutations at Positions C6S, W61Y, and S207T on Binding to NP

In order to prove the presence or absence of binding due to structural changes, each synthesized consensus gene was recombined with the pcDNA3.1-CMV vector to conduct an interaction study between NP, NSs, and NS mutants (Figure 2A). Simultaneously, we examined the interaction between NSs and NP by IP assay using HEK293T cells overexpressing each protein (Figure 2B). As can be seen from the results, the NP antibody-based IP fraction was detected in NS antibody-based immunoblotting, which indicates proper interaction between NP and original NS. Similarly, C6S and S207T mutants showed bands in NS antibody-based immunoblotting, indicating unaltered interaction of mutant NSs with NP. However, in the case of the W61Y mutant, no considerable band was detected, which implies that replacement of tryptophan at 61 with tyrosine affected the interaction (Figure 2B). Subsequently, the IFA assay demonstrated cytoplasmic co-localization of NSs and NP in HeLa cells transiently transfected with pcDNA3.1-NSs and NP or pcDNA3.1–NSsW61Y and NP (Figure 2C).

### 3.3. Comparing NSs and NS Mutant Structures

CD spectroscopy was performed to assess structural changes by mutation in Figure 3A. CD spectral data in mean residue ellipticity (deg cm^2^dmol^−1^) were analyzed by the online-based server CAPITO, and conformational changes were observed in the secondary structures as a result of NS mutation compared with the other reference proteins and NSs (Figure 3A). The protein structure was simulated, and intra-molecular interactions of wild-type and mutant NS proteins were analyzed in-silico. The structural analysis of C6S residue mutation has revealed no significant structural impact as it resides in the N-terminal loop of NSs. The predicted destabilization analysis (ΔΔG) of the C6S mutant has yielded −0.325 kcal/mol and no interatomic interaction has been observed. Whereas the mutation of Tyrosine in the 61st position of Tryptophan has significant impact in structural arrangement. Even though the W61Y mutation has induce the RMSD of 0.018 Å, the flip of the ring moiety and smaller in size has led to significant impact at the structural level. The mutation has been predicted to have the ΔΔG value of −0.086 kcal/mol. A total of four interatomic interactions were observed in W61 residue; Polar Van der Waals interaction at a distance 2.8 Å was observed between W61 and F63, the residue L177 has two Carbon-Pi interactions with W61 residue at the distance of 3.9 Å and 4 Å. In addition to that, hydrogen bond was observed between threonine 59 and W61 in the distance of 3.1 Å. Whereas in the case of Y61 mutations, only two interactions were observed (Polar Van der Waals interaction of F63 at the distance of 3 Å, and a carbon-Pi interaction of L177 at 3.88 Å). The comparative analysis revealed that mutation of Y61 has retained two interactions, however the flip has induced minimal difference in the interaction distances. Similar to W61Y mutation, the S207T has minimal level of structural changes with the RMSD of 0.017 Å. The mutation of S207T has made minimal level of displacement that lead to the differences in the atomic interaction distances. Among these three mutations, it is clearly evident that the W61Y mutation has higher structural impact, thus might have potential defect in biological function. The mutation exhibited destabilization from the wild-type NS structure with a ΔΔG value of about −0.139 kcal/mol. The S207Y mutant showed some minor changes in polar and weak polar van der Waals interactions, with a ΔΔG value of about −0.473 kcal/mol (Table 1). Meanwhile, the W61Y mutant showed considerable alteration in intra-molecular polar, H bond, van der Waals, and carbon-pi interactions. It experienced substantial destabilization, with a ΔΔG value of about −0.913 kcal/mol (Table 2; Figure 3B; Appendix A).

In order to further assess the role of mutations in biological assembly and function, we have performed protein-protein docking between NSs-NP and NS mutants-NP (Figure 3C). The docking scores of NSs and NSsC6S mutant with NP has revealed same docking score (−242.20 kcal/mol), which clearly depicts both the forms has no structural changes, and thus are not having any significant changes in biological action. The docking of W61Y mutant with NP has produced −175.92 kcal/mol as the resultant docking score, which comparatively lesser than among all other complexes. The conformational changes induced by the tyrosine mutation has led to the structural changes and resulted in lesser docking score. The docking of S207T mutant with NP had the score of −200.55 kcal/mol. The mutant threonine 207 residue present in hinge region has been playing a major role in domain movement, which is essential for protein-protein interactions. The docking score of S270T was better than that of the W61Y mutant, although minor conformational changes were observed. These data suggest that inhibition of the interaction between NP and NSsW61Y is due to structural changes.

### 3.4. Effects of W61Y NS Mutations on SFTSV Replication

Since, co-localization of NSs and NP was observed in the VLS (Figure 2) known as inclusion bodies (IBs) and it has been reported that VLS play a roles in viral replication [9,18]. Based on that, the effect of the NSW61Y mutation on viral replication was investigated. The transformant HeLa cells overexpressing wild-type and mutant NSs were subjected to qPCR assessment of S segment mRNA copy number and FAID_50_ after being infected with SFTSV. The qPCR result stated that a significant reduction in mRNA viral copy number, and a correlating result was observed in FAID_50_ determination (Figure 4A,B). Both results proved that decreased viral replication in the cell with exogenous expression of NSsW61Y in comparison to cells expressing wild-type NSs. These data showed that the mutation in NS suppressed viral replication, and thus implies the crucial role of NS in NS-NP interaction as well as in viral replication.

## 4. Discussion

The SFTSV S-segment is an ambisense RNA that encodes the NP and the NSs [19]. It has been reported that NS is mainly found in the cytoplasm and is associated with viral encapsidation [20]. The NS is a potentially important virulence factor that is involved in inhibition of host immune responses [21,22]. According to Wu et al., the NSs co-localize with NP from the S segment and form VLSs and IBs [9]. Based on 3D structural analysis of native NSs, there are three possible binding sites. Residues at 1 to 32, 56 to 82, and 207 to 237 amino acids may involve NS/NP interaction (Figure 1B). In order to understand the importance of NSs in interaction with NP and its influence in viral replication, we introduced point mutations by site-directed mutagenesis at each predicted binding site to determine the most actively participating site. Substitution of a particular amino acid residue with a larger residue can result in steric interference and should be avoided. Therefore, using residues smaller than the wild-type is preferable as they tend not to disrupt the protein’s overall structure. First, we replaced cysteine (C) at the sixth position with serine (S) as these two contain the most similar amino acids, only differing in a sulfur atom replacing the oxygen. Hence, the chain remains the same size and retains its hydrophilic properties. Next, we selectively changed tryptophan (W) at the 61st position with tyrosine (Y). Then, serine (S) at the 207th position was replaced with threonine (T). After substituting amino acids, ClusPro 2.0 was used to analyze the interaction between NS and NP (Figure 1C), and it showed altered interactions due to structural instability caused by amino acid variation. Previous studies have shown that NS forms a VLS, which is also known to be involved in viral replication of rotavirus [23]. The NP and NS were identified to co-localize in the VLS and to have interactions with each other [9]. Our co-immunoprecipitation results indicated that SFTSV NS was interacting with viral NP protein, which confirms the involvement of NSs-NP interaction in SFTSV replication (Figure 2). Interestingly, NSs mutated at W61Y failed to interact with the NP protein, which correlated with the results of the in-silico analysis that mutation of the 61st amino acid of the NSs caused substantial intra-molecular conformational changes, which played a major role in disrupting the binding to NP (Figure 3; Table 1). Furthermore, microscopic images showed that large portions of the NSs and NP were co-localized inside the VLSs (Figure 2C), and NP was dispersed mainly in the cell cytoplasm (Figure 2B,C). It is evident that NS proteins interact with NP and co-localize in VLSs. Therefore, it can be hypothesized that SFTSV replicates inside the VLSs, where the nucleocapsids are assembled and subsequently transported into the Golgi complex network to form the virions. Moreover, overexpression of NS protein in the HeLa cell line increased the SFTS viral copy number by five folds. Compared to the NSs, the NSsW61Y mutant less the viral replication about 1.9 folds (Figure 4). These comprehensive data suggest that NSsW61Y could not orchestrate the binding of endogenous viral NP, thus resulted in mitigated viral replication inside the VLS.

Numerous studies using Bunyamwera orthobunyavirus (BUNV) and Rift Valley fever (RVFV) have demonstrated that the NSs are involved in innate immunity by playing a role as viral interferon (IFN) antagonists. Unlike RVFV produced in the nucleus [24], SFTSV NSs are located in the cytoplasm and form VLS [9], which suppresses the interferon-beta promoter by binding with the TBK1/IKKε complex [8,25,26]. A previous study [27] stated that the NSs-A46 mutant could inhibit the IFN signal and induce IL−10 expression, which is critical in the immune evasion mechanism of SFTSV. Furthermore, the NSs-TRIM21 interaction-mediated activation of Nrf2 may have a specific role in viral pathogenesis. Moreover, NSs were sufficient to inhibit IFN-β induction by interfering with LSm14A as an activator of IFN regulatory factor3 (IRF3) [11]. Extracellular IFNs activate Janus kinase–signal transducer and activator of transcription and the phosphoinositide 3-kinase–AKT–mechanistic target of rapamycin pathway, which regulates the expression of autophagy-related genes. NSs directly interact with STAT2 and sequestrate it into NS granules [28]. However, the interaction of STAT1 with NSs is controversial [8,29], and the role of type II IFN in the anti-SFTSV function is yet to be elucidated. These collective findings of previous studies emphasize the significance of NS in viral pathogenesis. The current investigation provides a novel insight into the interaction between NS and NP, along with their co-localization. The SFTSV NP interact with viral RNA and plays a crucial role in the viral assembly [30]. The ribonucleoprotein (RNP) complex, composed of viral genomic RNA, N, and L proteins, serves as the molecular machinery for viral genome replication and transcription [10]. Thus, NP is directly involved in viral replication, while the present results suggest that NSW61 has an indirect impact on viral replication through its interaction with NP. This report offers vital information on the NS-NP interaction, which requires further investigation to identify potential targets against SFTSV effectively.

In conclusion, multiple bunyaviruses, including SFTSV, have been listed on the WHO R&D Blueprint, an international strategy aims to strengthen preparedness for future epidemics by urging the development of medical countermeasures, particularly vaccines and antivirals, which are currently limited [31]. To effectively devise antiviral strategies against emerging viruses like SFTSV and other bunyaviruses, a deep comprehension of the viral life cycle is imperative. This study demonstrated that NSsW61Y reduces the structural conformational integrity, which hampered viral replication through the NS-NP protein-protein interaction. In particular, the role of NSs in the interaction with NP and viral replication was demonstrated through in-silico structural analysis and docking study, in-vitro mutant protein expression, and protein interaction analysis. The roles of NS and its NP interaction were identified, and the mitigation of viral infectivity by the mutant NS was demonstrated. The present study emphasizes the critical site for NS-NP interaction, which plays a prominent yet indirect role in viral replication. This can be studied further to develop therapeutic regimens to arrest the NS-NP interaction and dependent viral propagation.

## Figures and Tables

**Figure 1 viruses-15-01963-f001:**
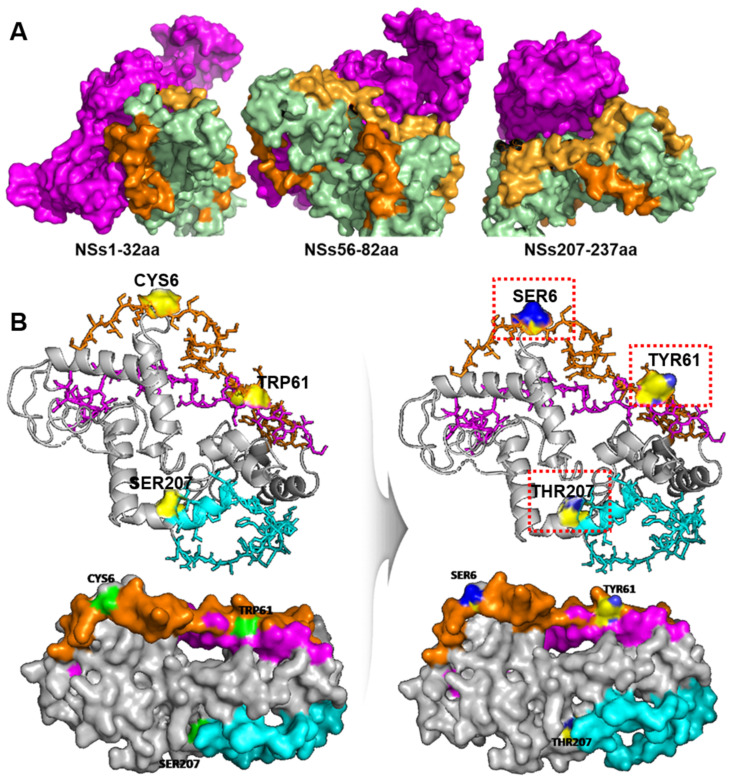
Structure characterization of the NP, NSs, and NS mutants, and schematic diagram describing the generation of recombinant vectors. (**A**) Predicted 3D structures representing NS (bottom) and NP (violet, top) binding sites at amino acid residues from 1 to 32, 56 to 82, and 207 to 237. (**B**) The 3D structural features of NS mutants. The orange regions highlight the interacting amino acids of the STSV NS protein, and the blue regions highlight the replaced amino acids after site-directed mutagenesis (light panel, red box).

**Figure 2 viruses-15-01963-f002:**
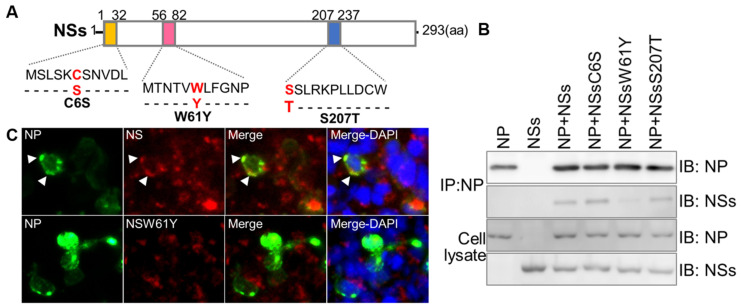
Effects of NS protein mutations at positions C6S, W61Y, and S207T on binding to NP. (**A**) Schematic diagram pcDNA3.1-CMV plasmid constructs of the harboring NP, NSs, and NSs mutants. (**B**) HEK293T cells were co-transfected with plasmids of pcDNA3.1-NSs, pcDNA3.1-NSsW61Y, and pcDNA3.1-NP and collected at 48 h. Interaction between NP and NSs constructs was detected by immunoprecipitation (IP). (**C**) HeLa cells were co-transfected with pcDNA3.1-NP, pcDNA3.1-NSs, and pcDNA3.1-NSsW61Y. After 48 h, co-transfected cells were fixed with 80% cold acetone and stained with anti-rabbit-NP and anti-mouse-NS primary antibodies. The co-localization of each protein was visualized with alexa488-anti-rabbit IgG (green) and alexa647-anti-mouse IgG (red), using a secondary antibodies fluorescence microscope (×100). Nuclei were stained with DAPI (blue).

**Figure 3 viruses-15-01963-f003:**
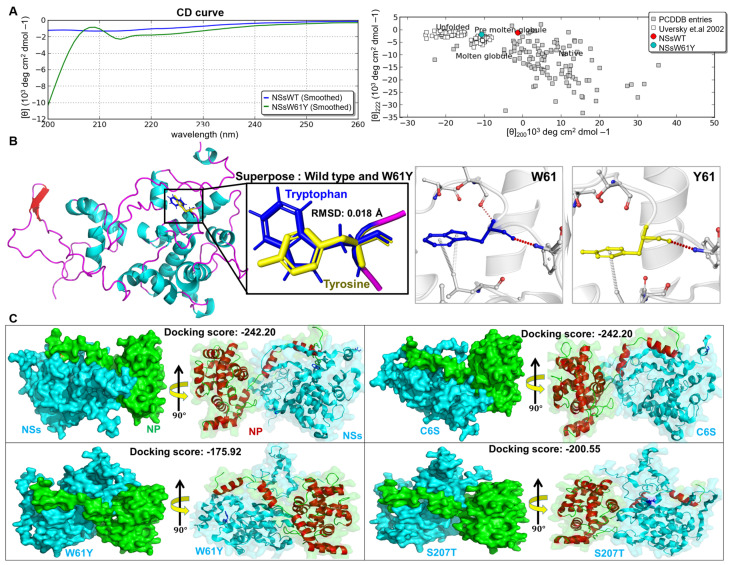
The 3D structures of SFTSV NSs and W61Y from two views. (**A**) Structural alterations in NSs and NSW61Y as measured by CD spectroscopy. The secondary structure of NS and NSW61Y were analyzed by CD spectroscopy. The spectral change in millidegree and mean residual ellipticity are plotted against wavelength and presented graphically. The mean residual ellipticity [Ɵ] was interpreted by an online based server CAPITO. (**B**) The effect of W61Y alteration in the interaction with surrounding residues. Simulated interaction of NSsW61Y with NP protein shows H-bond, van der waals binding differences compared to wild types. (**C**) The tri-dimensional docking structure of NSs and NSs mutants (cyan) interaction with NP (green and red) protein shows representative surface (**left**) and cartoon (**right**) structure, respectively. The docking score has changed due to local conformational changes induced by the mutant residue.

**Figure 4 viruses-15-01963-f004:**
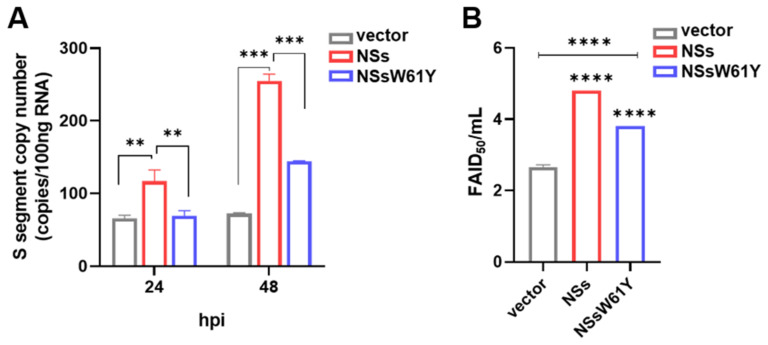
Role of NSs in SFTSV replication. (**A**) HeLa cells were transfected with pcDNA3.1 (+), pcDNA3.1-NS, and pcDNA-NSW61Y plasmids. At 48 h post-transfection, cells were infected with SFTSV at MOI of 0.5 for 24 h and 48 h, and relative mRNA expression was measured by qRT-PCR. (**B**) After 48 h, Cell culture media were taken for virus titration and titers were measured by FAID_50_. The one-way ANOVA comparison test was used to assess *p*-values by the two-tailed, unpaired *t*-test. *p*-values indicate statistical significance (** *p* ≤ 0.01, *** *p* < 0.001, and **** *p* < 0.0001).

**Table 1 viruses-15-01963-t001:** Interaction analysis.

Tyrosine (Y61) Interacts with Residue	Interaction Type	Distance (Å)
Phenylalanine (F63)	Polar Van der Waals	3
Leucine (L177)	Carbon-Pi	3.8
**Tryptophan (W61)** **interacts with residue**		
Phenylalanine (F63)	Polar Van der Waals	2.8
Leucine (L177)	Carbon-Pi	3.9
Leucine (L177)	Carbon-Pi	4
Threonine (59)	H-bond Van der Waals	3.1
**Serine (S207)** **interacts with residue**		
Threonine (T205)	Polar	3.2
Arginine (R210)	Weak Polar Van der Waals	3.3
Arginine (R210)	Weak polar	3.5
Arginine (R210)	Weak polar	3.4
**Threonine (T207)** **interacts with residue**		
Threonine (T205)	Polar	3.3
Arginine (R210)	Weak Polar Van der Waals	3.0
Arginine (R210)	Weak h-bond Van der Waals	3.0
Arginine (R210)	Weak Polar Van der Waals	3.3

**Table 2 viruses-15-01963-t002:** List of predicted binding energy.

	NSsC6S	NSsW61Y	NSsS207T
**RMSD (Å)**	0.016 (Destabilizing)	0.018 (Destabilizing)	0.017 (Destabilizing)
**Binding energy** **(Kcal/Mol)**			
**ΔΔG**	−0.325	−0.086	−0.913

## Data Availability

Data are available from the authors upon reasonable request.

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
