# Peer review of "Non-Structural Protein-W61 as a Novel Target in Severe Fever with Thrombocytopenia Syndrome Virus (SFTSV): An In-Vitro and In-Silico Study on Protein-Protein Interactions with Nucleoprotein and Viral Replication"

_viruses, 2023, doi:10.3390/v15091963_

Round 1

Reviewer 1 Report

Dear authors,

Your manuscript “Unveiling the Interactions of Non-structural Protein-W61 with nucleoprotein: and as a Potential Therapeutic Target in Severe Fever with Thrombocytopenia Syndrome Virus (SFTSV) Replication” describes investigations of the significance of the association between NS and NP during viral infection through in-silico and in-vitro analyses.

 However, in current form the manuscript needs the revision.

First, I recommend changing the manuscript title, in current form it is somewhat embarrassing.

 At abstract section, it is not clear without reading the entire text what the abbreviations mean, in my opinion, it should also be rewritten.

 Lane 36 – “encoding RdRp” – RdRp needs to be stated.

 Lane 90 – “E. coli” – should be in italic.

 General comment:

It is not clear, you compared the properties of recombinant proteins and their mutant forms obtained from E.coli, but to what extent did these recombinant proteins correspond in their properties to native ones, because native and recombinant proteins can differ significantly in their properties? Have you done such research?

Author Response

Dear Editor,

We want to thank you and the reviewers for the constructive comments made on our manuscript that have overall improved our manuscript substantially. We have addressed all the comments and issues raised by the reviewer and hope you will find our revised manuscript suitable for publication in your reputed journal. All the changes have been marked in blue in the revised manuscript.

Reviewer: 1

Comment 1. TitleI recommend changing the manuscript title, in current form it is somewhat embarrassing 

Response: As per the reviewer’s comment, appropriate title has been incorporated in the revised manuscript. (Page: 1, Line: 2-4)

Comment 2. Abstract: At abstract section, it is not clear without reading the entire text what the abbreviations mean, in my opinion, it should also be rewritten.

Response: As per the reviewer’s comment, appropriate corrections were made for better clarity and independent understanding of the abstract in the revised manuscript. (Page: 2, Line: 29-31)

Comment 3. Lane 36 : encoding RdRp” – RdRp needs to be stated.

Response: As per the reviewer’s comment, appropriate details were incorporated in the revised manuscript. (Page:3, Line: 49)

Comment 4. Lane 90 : E. coli” – should be in italic.

Response: As per the reviewer’s comment, appropriate corrections were incorporated in the revised manuscript. (Page: 5, Line: 102)

Comment 5. General comment

It is not clear, you compared the properties of recombinant proteins and their mutant forms obtained from E.coli, but to what extent did these recombinant proteins correspond in their properties to native ones, because native and recombinant proteins can differ significantly in their properties? Have you done such research?

Response: We agree with the reviewer’s opinion regarding the different properties of native and recombinant proteins obtained from E. coli. 

Our aim is to understand the possible structural and functional alterations caused by the point mutations. To acquire a preliminary understanding, we opt to express and purify the naïve NS and mutant NSs proteins in the E. coli system and observed alteration in circular dichroism results.

As an answer to the reviewer’s query and to understand the real scenario, we also expressed the naïve NS and mutant NSs proteins through the pcDNA3.1 plasmid in eukaryotic host cells. The proteins expressed in this way would be very similar to the original viral proteins in terms of structural properties, as we use appropriate host cell lines.  

Overall, we have done preliminary analysis using E. coli expressed proteins, and for deeper confirmation, Immunofluorescence assay (IFA), fluorescence active infectious dose (FAID50) assay, and Immunoprecipitation (IP) assays were conducted via eukaryotic protein expression. 

Again, we sincerely appreciate your constructive comments. We trust that our revisions have addressed all the questions and hope that you will find the revised manuscript acceptable for publication in the Pharmaceutics.

We truly appreciate your reviewing the manuscript.

Sincerely yours,

John Hwa Lee, D.V.M., Ph.D

Professor, Jeonbuk National University, 

College of Veterinary Medicine, 

Iksan Campus, 54596, Republic of Korea

Tel.: +82-63-850-0940. 

E-mail: [email protected].

Reviewer 2 Report

In this manuscript, Ji-Young et al., through their in-silico structural analysis and in vitro NSs and NP protein-protein interaction, decipher the critical binding site of SFTSV NS and its interaction with NP. Importantly, they found a crucial site in NSs-NP interaction, which plays an essential role in SFTSV viral replication. The manuscript is well-written, with quite convincing data. However, some improvements should be revised in this manuscript.

1. The methodology part should be as detailed as possible, such as the concentration of antibodies used by the author for immunoprecipitation IP) assay and WB analysis and how to evaluate the consistency of the titers of the immune antibodies used to make the comparisons meaningful. SDS-PAGE for protein purity is essential. Mainly, these anti-serums are used for Co-IP analysis.

2. The authors selected 12 strains of SFTSV to generate the consensus sequences. The amino acid alignment sequence should be displayed to make it more transparent and more accessible for readers to understand.

3. In Supplementary Table 1, please check it carefully and make it look more consistent. e.g. Strains/plasmid, pcDNA3.1-CMV, pcDNA-NP, not 3.1 version? Only express three E. coli proteins?

4. In line 150, the author mentioned immunoblotting as described above, but can not find any information.

5. In Figure 2B, please label more precise, every lane.

Author Response

Dear Editor,

We want to thank you and the reviewers for the constructive comments made on our manuscript that have overall improved our manuscript substantially. We have addressed all the comments and issues raised by the reviewer and hope you will find our revised manuscript suitable for publication in your reputed journal. All the changes have been marked in blue in the revised manuscript.

Comment 1. The methodology part should be as detailed as possible, such as the concentration of antibodies used by the author for immunoprecipitation IP) assay and WB analysis and how to evaluate the consistency of the titers of the immune antibodies used to make the comparisons meaningful. SDS-PAGE for protein purity is essential. Mainly, these anti-serums are used for Co-IP analysis. 

Response: Detailed description has been included in the materials and method part of the revised manuscript. (Page: 7-8, Line: 155-189, 162-169)

Comment 2. The authors selected 12 strains of SFTSV to generate the consensus sequences. The amino acid alignment sequence should be displayed to make it more transparent and more accessible for readers to understand. 

Response: As per the reviewer’s comment, the aligned consensus sequences were attached the supplementary data 2 and 3. (Page: 5, Line: 90-91)

Comment 3. Supplementary Table 1:  please check it carefully and make it look more consistent. e.g. Strains/plasmid, pcDNA3.1-CMV, pcDNA-NP, not 3.1 version? Only express three E. coli proteins?

Response: As per the reviewer’s comment, appropriate details were incorporated consistently in the revised supplementary Table 1. (Page:1). Since three E. coli proteins were intended to produce hyperimmune sera for antibodies and CD analysis, and NSsW61Y variant protein was also purified by E. coli expression system. 

Comment 4. line 150 : the author mentioned immunoblotting as described above, but can not find any information.

Response: As per the reviewer’s comment, details were incorporated in the revised manuscript. (Page: 8, Line: 162-169)

Comment 5. Figure 2B:  please label more precise, every lane.

Response: As per the reviewer’s comment, figure was incorporated in the revised manuscript. (Page: 26) 

Again, we sincerely appreciate your constructive comments. We trust that our revisions have addressed all the questions and hope that you will find the revised manuscript acceptable for publication in the Pharmaceutics.

We truly appreciate your reviewing the manuscript.

Sincerely yours,

John Hwa Lee, D.V.M., Ph.D

Professor, Jeonbuk National University, 

College of Veterinary Medicine, 

Iksan Campus, 54596, Republic of Korea

Tel.: +82-63-850-0940. 

E-mail: [email protected].

Round 2

Reviewer 2 Report

The author has revised the manuscript accordingly, I have no further comments.